

# Adaptation and validation of the Chinese version of the Hospice Comfort Questionnaire-Patient (HCQ-P)

Nana Xu[1,2], Xu Yan[1], Xiaohong Ou[3], Jun Ren[2] and Qiaoqin Wan[1]

[1] Peking University, School of Nursing, Beijing, China
[2] Chinese PLA General Hospital, The Second Medical Center & National Clinical Research Center for Geriatric Diseases, Beijing, China
[3] Beijing Haidian Hospital, Palliative Care Unit, Beijing, China

Corresponding author
Qiaoqin Wan, qqwan05@163.com

## ABSTRACT

**Objective**. The Hospice Comfort Questionnaire-Patients (HCQ-P) is widely used to assess patient's comfort levels in hospice care. This research aimed to culturally adapt the HCQ-P for the Chinese context and validate its psychometric properties to ensure its applicability and effectiveness in China.

**Methods**. This research was conducted in two main phases: (1) translation and cross-cultural adaptation of the HCQ-P into Chinese, (2) evaluation of the psychometric properties through expert consultations and a cross-sectional survey among 360 hospice care patients. The evaluation included determining floor and ceiling effects, evaluating internal consistency using Cronbach's $\alpha$, and testing test-retest reliability with the intra-class correlation coefficient (ICC). Content validity was assessed using the content validity index (CVI), and construct validity was tested through confirmatory factor analysis (CFA).

**Results**. The HCQ-P was successfully translated and culturally adapted into Chinese, with no significant floor or ceiling effects detected. The overall Cronbach's $\alpha$ for the HCQ-P was 0.94, demonstrating excellent internal consistency, while dimension-specific alphas ranged from 0.77 to 0.84. The overall ICC was 0.93, indicating high test-retest reliability, with individual dimensions ranging from 0.77 to 0.81. Both item-level and scale-level CVIs reached 1, reflecting unanimous expert agreement on content relevance. Significant factor loadings in the CFA confirm that the HCQ-P is statistically robust and well-aligned with the cross-cultural and clinical contexts of Chinese hospice care.

**Conclusions**. The Chinese version of the HCQ-P exhibits robust psychometric properties, making it a valid and reliable instrument for assessing patient comfort in Chinese hospice care settings.

## INTRODUCTION

Hospice care is a patient-centered approach that aims to enhance comfort and preserve dignity during the end-of-life period. Introduced by *Saunders (1978)*, it emphasizes pain relief, symptom management, and psychospiritual and spiritual support to help patients

face their final stages with dignity while providing emotional support to families and caregivers. As global awareness of quality of life grows, hospice care is increasingly accepted and expanded to include various stages of treatment for serious chronic diseases. According to the WHO, approximately 40 million people need palliative care each year, with 78% living in low- and middle-income countries, but only about 14% have access to such services. The demand for palliative care is expected to double by 2060. In China, hospice care has advanced since the release of the Guidelines for Hospice Care Practice (Trial) in 2017, forming a diversified service model. However, it remains in its infancy with incomplete national legislation, though the 2023 launch of the third batch of hospice care pilot projects marks a new phase toward national expansion.

Comfort in hospice care is considered both a personalized and holistic experience, serving as a source of patient satisfaction and well-being, and plays a vital role in high-quality care (*Trotte & Caldas, 2015*). It reflects both the individualized subjective experience and the quality of healthcare. Various comfort assessment tools have been developed globally to address different populations, covering patients, professional caregivers, and family caregivers. Comfort assessment tools for patients include observer-rated scales and self-reported scales.

Among observer-rated tools, the Comfort Scale (CS) (*Ambuel et al., 1992*) simulates ICU nurses' clinical judgment regarding patient distress. The Comfort Behavior Scale (CBS), a revision of the CS by *Carnevale & Razack (2002)*, was initially used to assess sedation in pediatric resuscitation for intubated children (*Carnevale & Razack, 2002*; *Ista et al., 2005*) and later applied widely in pediatric intervention evaluations (*Koopman et al., 2018*; *Hazwani et al., 2022*; *Liu & Ge, 2019*; *Bai et al., 2012*). The End-of-Life Dementia Comfort Assessment Scale (EOLD-CAD) evaluates symptom control in dementia patients at the end of life (*Volicer, Hurley & Blasi, 2001*), and was validated by *Kiely et al. (2006)* and *Yeh et al. (2021)* for hospice care interventions. For self-reported tools, the Visual Analog Scale (VAS) was initially designed to assess pain intensity (*Woodforde & Merskey, 1972*) but is widely used to evaluate subjective comfort experiences (*Li, Liu & Herr, 2007*; *Bozdemir et al., 2022*), though limited to a single dimension. Other scales, like the Subjective Well-being Scale (SWN) (*Naber et al., 2001*) and the Patient Evaluation of Emotional Comfort Experienced (PEECE) (*Williams et al., 2017*), have limited target audiences. The General Comfort Questionnaire (GCQ), developed by *Kolcaba (1992)*, evaluates comfort from both dimensional and hierarchical perspectives and has been widely applied due to its high reliability and validity (*Góis et al., 2018*; *Vicdan, 2020*; *Hu, Xu & Xu, 2023*). For professional caregivers, comfort assessment tools focus on emotional experiences in nursing practice. The Nurse Comfort Questionnaire (NCQ) (*Cinar Yucel et al., 2019*) assesses nurses' comfort in end-of-life care, correlating it with care quality. The Comfort with Communication in Palliative Care Scale (C-COPE) (*Isaacson & Minton, 2018*) evaluates healthcare providers' comfort in communication during hospice care (*Styes & Isaacson, 2021*; *Wittenberg et al., 2022*) and has been applied in Chinese studies (*Ji et al., 2023*). For family caregivers, the Family Caregiver Comfort in Critical Care Scale (ECONF) (*Freitas, Menezes & Mussi, 2015*) evaluates their sense of security, social support, and family interaction.

Despite these tools, the field faces challenges. Internationally, the Hospice Comfort Questionnaire (HCQ), adapted from the GCQ, is preferred for assessing the comfort of hospice patients and caregivers (*Novak et al., 2001*). It includes two subscales: the Hospice Comfort Questionnaire-Patient (HCQ-P) and Hospice Comfort Questionnaire-Caregiver (HCQ-C), covering physical, psychospiritual, sociocultural, and environmental dimensions. Designed for hospice populations, the HCQ is accurate and reliable, addressing patients' unique end-of-life needs, such as pain management and symptom relief. International applications show it is effective in guiding medical decisions and interventions. In contrast, hospice care comfort assessment in China is relatively underdeveloped, relying on the Chinese version of the GCQ (*Bian, 2022*; *Mei et al., 2021*; *Zhao, Huang & Jin, 2020*). There is no universally recommended tool for evaluating hospice patient and caregiver comfort (*Lorente, Losilla & Vives, 2018*). Thus, there is an urgent need for a comprehensive, targeted assessment tool in China. Moreover, research on comfort factors in hospice care is limited, and a holistic approach is needed to analyze factors affecting both patients' and caregivers' comfort, to improve overall care.

In response, this study aims to introduce the Hospice Comfort Questionnaire-Patients (HCQ-P) to China, conducting translation, cultural adaptation, and psychometric validation. The study will also integrate cross-sectional surveys and literature reviews to identify key factors influencing the comfort of hospice patients and caregivers, ultimately providing a precise comfort assessment tool to improve hospice care quality and optimize intervention strategies.

## METHODS

In this research, we utilized the HCQ-P adapted from the GCQ. The HCQ-P encompasses four dimensions of comfort: physical, psychospiritual, Sociocultural, and environmental, structured into 49 items scored on a six-point Likert scale ranging from strongly disagree to strongly agree. The total scores vary from 49 to 294, with higher scores indicating greater comfort levels. Specifically, items 2, 5, 6, 12, 13, 14, 17, 19, 21, 22, 24, 25, 26, 27, 30, 32, 34, 38, 39, 40, 43, 45, and 48 are reverse-scored to ensure that a higher numerical response consistently reflects a higher comfort level. The research was executed in two phases (see Fig. 1): (1) Translation and cross-cultural adaption, and (2) Psychometric evaluation. Ethical approval was obtained from Peking University biomedical ethics committee (Approval No. IRB00001052-24038).

### Phase 1: Translation and cross-cultural adaption

To adapt and validate the HCQ-P into Chinese, a systematic translation and cultural adaptation process was conducted in accordance with established guidelines after getting permission from the HCQ-P developer. The detailed steps were as follows:

### *Step 1: Forward translation*

Two bilingual translators, both postgraduates majoring in nursing and fluent in English and Chinese, independently translated the original HCQ-P into Chinese. This step resulted in two initial forward translations (FT1 and FT2).

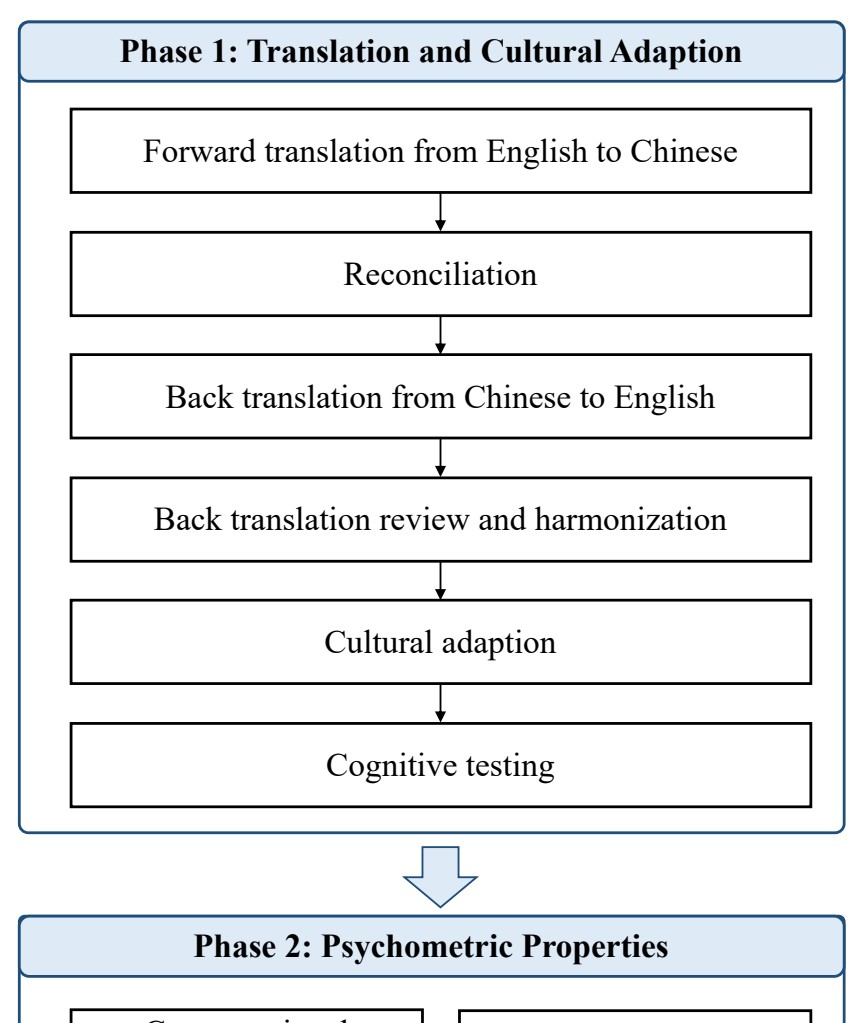

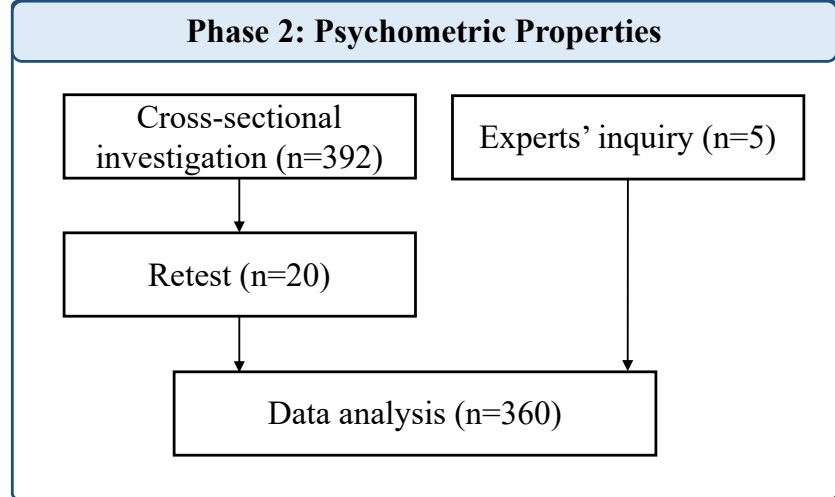

**Figure 1** Flowchart of the translation and cross-cultural adaptation process of HCQ-P from the original English version.

### Step 2: Reconciliation

The two forward translations were compared and reconciled by the original translators and a third independent bilingual translator with expertise in medical terminology. Discrepancies were resolved through discussion, resulting in a consolidated forward translation (FT12).

### Step 3: Back translation

Two professional translators, both native English speakers with advanced knowledge of Chinese and experience in healthcare-related translation, independently back-translated FT12 into English. These translators had not been exposed to the original HCQ-P. The back translation step yielded two versions (BT1 and BT2).

### Step 4: Back translation review and harmonization

A review panel, including the research team and the translators involved, was convened to compare BT1 and BT2 with the original HCQ-P. Conceptual and semantic equivalence were assessed, and any discrepancies were clarified through consultation with the original HCQ-P developer. After harmonization, a pre-final version A of the Chinese HCQ-P was established.

### Step 5: Cultural adaption

The pre-final version A underwent a cultural adaptation process. A panel of eight experts, including hospice care professionals and cross-cultural translation specialists, was invited to evaluate the semantic and cultural equivalence of the pre-final version A. Experts were selected based on the following criteria: (1) at least 5 years of experience in hospice care or cross-cultural research, (2) fluency in English and Chinese, and (3) willingness to participate. Feedback was collected through a four-point Likert scale (ranging from 1 = not applicable to 4 = totally applicable) and open-ended suggestions. Based on their feedback, modifications were made to form a pre-final version B.

### Step 6: Cognitive testing

Pre-final version B was tested through cognitive interviews with 12 clinical professionals (six physicians and six nurses) who were actively engaged in hospice care. Participants completed the pre-final version B and were subsequently interviewed regarding their comprehension and perceptions of each item. Their feedback guided the refinement of the scale, leading to the finalized Chinese version of the HCQ-P.

## Phase 2: Psychometric evaluation

The psychometric evaluation of the Chinese version of HCQ-P focused on its validity (content and construct validity) and reliability (internal consistency and test–retest reliability). Ethical approval was obtained from Peking University biomedical ethics committee (Approval No. IRB00001052-24038). This phase was divided into two components: expert consultations for content validity and a cross-sectional survey to evaluate other psychometric properties.

### Experts' inquiry

Expert consultations were conducted to evaluate the content validity of the HCQ-P. Inclusion criteria for experts were as follows: (1) having extensive experience in hospice care practice and research; (2) having at least five-year professional experience; (3) willing to participate in the study. Finally, three experts were invited to assess the relevance of each item in the HCQ-P to the underlying measurement objectives. Evaluations were performed using a four-point Likert scale (1 = Not at all related to 4 = Very related). Feedback from the experts was used to calculate the item-level content validity index (I-CVI) and scale-level content validity index (S-CVI) for the HCQ-P.

### Cross-sectional survey

*Participants.* Convenience sampling was used due to the practical limitations in accessing hospice patients, which is a commonly adopted method in palliative care research (*Etikan, Musa & Alkassim, 2016*). From December 2023 to June 2024, patients were recruited from hospice care wards and oncology wards. Inclusion criteria for participants were: (1) receiving hospice care services in outpatient or inpatient settings; (2) capable of subjective communication; (3) physically and cognitively able to complete the questionnaire independently or with assistance; and (4) provided informed consent and voluntarily agreed to participate. Informed consent was obtained in written form from all participants. Patients younger than 18 years or experiencing severe health changes within one week were excluded from the study.

*Sample size.* According to the Kendall sample estimation method, the sample size should be 5–10 times the number of questionnaire items (*Shoukri, Asyali & Donner, 2004*). Given that the HCQ-P has 49 items, the minimum sample size was set at 245 participants. To account for potential non-responses, the sample size was increased by 20%, at least 307 participants were required for confirmatory factor analysis (CFA) to ensure adequate statistical power. The 20% adjustment is a widely recognized practice in survey-based research to account for non-response bias and enhance the reliability of statistical analysis (*Krejcie & Morgan, 1970*; *Shoukri, Asyali & Donner, 2004*). While sensitivity analyses were not conducted in this study due to time and resource constraints, future research could incorporate such analyses to further evaluate the robustness of sample size assumptions.

*Data collection.* The pre-survey phase took place from December 2023 to January 2024, followed by the formal survey from February 2024 to June 2024. After obtaining informed consent, research team members distributed questionnaires with standardized instructions through the Questionnaire Star by sending the website link or QR code to the potentially eligible participants. The questionnaire consisted of three sections: (1) instructions and informed consent to explain the research's purpose, principles of anonymity, and voluntary participation; (2) demographic characteristics including gender, age, education level, marital status, primary diagnosis, treatment history, and experience with hospice comfort services; (3) Chinese version of the HCQ-P. Finally, a total of 392 questionnaires were distributed, and 360 valid responses were received, yielding a response rate of 91.8%. For test–retest reliability, we sent HCQ-P again after a three-week interval, and 20 valid

questionnaires were received. The data that support the findings of this study are available from the supplemental files.

### Data analysis

Demographic characteristics were presented by frequencies and percentages. The scores of the HCQ-P were presented with means and standard deviations. All analyses were conducted using the SPSS 27.0 and Amos 24.0 software.

*Floor and ceiling effects.* Floor and ceiling effects were evaluated by calculating the minimum or maximum socres on the HCQ-P. Floor or ceiling effects were considered absent if less than 15% of participants achieved the lowest or highest scores (*Terwee et al., 2007*).

*Internal consistency reliability.* Internal consistency reliability was evaluated by calculating Cronbach's α coefficients for each dimension and the total scale. Cronbach's α coefficients ≥ 0.70 were considered satisfactory (*Mokkink et al., 2016*). Additionally, item-total correlation coefficients were calculated to measure the relationship between individual items and the total score. Items with a correlation coefficient < 0.20 were considered for deletion (*Zhang, Ge & Rask, 2019*).

*Test–retest reliability.* To assess the stability of the questionnaire over time, the intraclass correlation coefficient (ICC) was calculated using Spearman's Rho correlation analysis. An ICC value ≥ 0.70 indicated good test–retest reliability, while 0.60 was acceptable (*De Vet et al., 2006*; *Jöreskog, 1971*).

*Content validity.* The content validity index (CVI) was assessed at both the item level (I-CVI) and scale level (S-CVI) based on expert ratings. I-CVI was calculated as the proportion of experts rating each item as 3 (related) or 4 (very related) on a four-point Likert scale. The S-CVI was computed as the average of all I-CVIs. Content validity was considered satisfactory if I-CVI ≥ 0.78 and S-CVI ≥ 0.80 (*Lynn, 1986*).

*Construct validity.* Construct validity was evaluated using confirmatory factor analysis (CFA) to test the original four-factor structure of the HCQ-P. Model fit was assessed using the following indices: $\chi^2$/degrees of freedom ratio ($\chi^2$/df < 3.00 indicated good, while < 0.5 was acceptable), root mean square error of approximation (RMSEA < 0.08), goodness of fit index (GFI > 0.90), Tucker–Lewis index (TLI > 0.90), and comparative fit index (CFI > 0.90) (*Worthington & Whittaker, 2006*). Item factor loadings were also examined, with a cutoff of 0.40 used to retain items in the model (*McNeish, An & Hancock, 2018*).

## RESULTS

### Phase 1: Translation and cross-cultural adaptation process

Translation discrepancies primarily emerged during the back translation review and harmonization (Step 4) and the cultural adaptation (Step 5). The expert panel suggested changes to enhance clarity and cultural relevance. The first issue is that the expression of

physical discomfort varied significantly across different diseases, prompting a refinement of terminologies used to describe bodily sensations in the Chinese context. After discussion in the harmonization meeting and communication with the developer, we modified item 5 from 'I feel bloated' to 'I feel my body swelling'. Additionally, the distinction between environmental, psychospiritual and physical comfort became crucial as certain items translated directly could lead to ambiguity. For instance, item 32 was changed from 'This chair (bed) makes me hurt' to 'This chair (bed) makes me uncomfortable', a more specific expression to reflect environmental discomfort caused by furniture, and item 38 from 'I feel out of place here' to 'I'm not comfortable here' to better convey the sense of environmental discomfort. Finally, 12 participants reviewed the Chinese HCQ-P for clarity and relevance. All confirmed the items were well-articulated and understandable, necessitating no revisions. The items from both the original English and final Chinese versions are detailed in Table 1, demonstrating effective adaptation for Chinese hospice care contexts.

## Phase 2: Psychometric evaluation
### Participant characteristics
A total of 392 questionnaires were collected from hospice care wards and oncology wards. After excluding ineligible and obviously invalid responses, 360 valid questionnaires were included in the analysis, resulting in an effective response rate of 91.8%. There were no missing data, as each item in the electronic questionnaire was mandatory. Detailed demographic characteristics of the participants are provided in Table 2, while the HCQ-P item scores are presented in Table 3.

### Floor and ceiling effects
In the validation of the HCQ-P, with a scoring range from 49 to 294, the observed highest and lowest scores were 280 and 144, respectively, each recorded by two participants (0.56%). Given that floor and ceiling effects are significant if more than 15% of participants score at these extremes, the minimal occurrence indicated no floor or ceiling effects in HCQ-P.

### Internal consistency reliability
The overall reliability of the HCQ-P was reflected in a high Cronbach's α of 0.94. When analyzing the subscales, the reliability coefficients varied, with the physical Comfort Scale scoring the highest at 0.841 and the Sociocultural Comfort Scale the lowest at 0.772 (see Table 4). Notably, all item-total correlation coefficients were statistically significant ($p < 0.001$) and the absolute value were greater than 0.2 except for item HCQ-P26, I would like to see my doctor more often, which is part of the Sociocultural Comfort Scale. Furthermore, prior to its exclusion, the Cronbach's α coefficient of Sociocultural Comfort would increase. Therefore, item 26 was excluded (see Table 3).

### Test–retest reliability
The Intraclass Correlation Coefficient (ICC) for the HCQ-P showed excellent stability with an overall ICC of 0.93. The ICCs for individual dimensions ranged from 0.772 in Sociocultural Comfort to 0.812 in physical Comfort, confirming the Chinese version of

**Table 1   The items of the HCQ-P in both English version and Chinese version.**

| Items | English version | Chinese version |
|---|---|---|
| HCQ-P1 | My body is relaxed right now | 我的身体现在很放松 |
| HCQ-P2 | My breathing is difficult | 我呼吸困难 |
| HCQ-P3 | I have enough privacy | 我有足够的隐私 |
| HCQ-P4 | There are those I can depend on when I need help | 当我需要帮助时，有人可以依靠 |
| HCQ-P5 | I feel bloated | 我感到身体肿胀 |
| HCQ-P6 | I worry about my family | 我担心我的家庭 |
| HCQ-P7 | My beliefs give me peace of mind | 我的信仰使我内心平和 |
| HCQ-P8 | My nurse(s) give me hope | 护士给了我希望 |
| HCQ-P9 | My life is worthwhile right now | 我的生命是有价值的 |
| HCQ-P10 | I know that I am loved | 我知道有人爱我 |
| HCQ-P11 | These surroundings are pleasant | 周围的环境令人愉悦 |
| HCQ-P12 | I have difficulty resting | 我难以放松 |
| HCQ-P13 | No one understands me | 没有人理解我 |
| HCQ-P14 | My pain is difficult to endure | 我的疼痛难以忍受 |
| HCQ-P15 | I feel peaceful | 我感到平静 |
| HCQ-P16 | I sleep soundly | 我睡得很安稳 |
| HCQ-P17 | I feel guilty | 我感到愧疚 |
| HCQ-P18 | I like being here | 我喜欢在这里 |
| HCQ-P19 | I am nauseated | 我感到恶心 |
| HCQ-P20 | I am able to communicate with my loved ones | 我能够与我爱的人交流 |
| HCQ-P21 | This room makes me feel scared | 我能够与我爱的人交流 |
| HCQ-P22 | I am afraid of what is next | 这个房间让我感到害怕 |
| HCQ-P23 | I have special person(s) who make(s) me feel cared for | 我害怕接下来会发生的事情 |
| HCQ-P24 | I have experienced changes which make me feel uneasy | 生活中有特别关心我的人 |
| HCQ-P25 | I like my room to be quiet | 我经历过让我感到不安的变化 |
| HCQ-P26 | I would like to see my doctor more often | 我喜欢我的房间是安静的 |
| HCQ-P27 | My mouth and skin feel very dry | 我想更频繁地见到医生 |
| HCQ-P28 | I'm okay with my personal relationships | 我感到嘴巴和皮肤非常干燥 |
| HCQ-P29 | I can raise above my pain | 我可以克服疼痛 |
| HCQ-P30 | The mood around here is depressing | 这里气氛压抑 |
| HCQ-P31 | I am at ease physically | 我现在身体是放松的 |
| HCQ-P32 | This chair makes me hurt | 我坐的椅子让我不舒服 |
| HCQ-P33 | This view inspires me | 这里的景象激励着我 |
| HCQ-P34 | I think about my discomforts constantly | 我总是在想我的不舒适 |
| HCQ-P35 | I feel confident spiritually | 我在精神上感到自信 |
| HCQ-P36 | I feel enough to do some things for myself | 我认为自己能够为自己做一些事情 |
| HCQ-P37 | My friends remembers me with their cards and phone calls | 我的朋友惦记着我，给我寄卡片、打电话 |
| HCQ-P38 | I feel out of place here | 我在这里感到不自在 |
| HCQ-P39 | I need to be better informed about my condition | 我需要更好地了解我的情况 |

**Table 1** (*continued*)

| Items | English version | Chinese version |
|---|---|---|
| HCQ-P40 | I feel helpless | 我感到无助 |
| HCQ-P41 | My god is helping me | 老天在帮助我 |
| HCQ-P42 | This room smells fresh | 这个房间空气清新 |
| HCQ-P43 | I feel lonely | 我感到孤独 |
| HCQ-P44 | I am able to tell people what I need | 我能够告诉别人我的需求 |
| HCQ-P45 | I am depressed | 我感到抑郁 |
| HCQ-P46 | I have found meaning in my life | 我找到了人生的意义 |
| HCQ-P47 | In retrospect, I've had a good life | 回顾过去，我这辈子很不错 |
| HCQ-P48 | My loved ones' state of mind makes me feel sad | 我所爱的人的心态让我感到悲伤 |
| HCQ-P49 | The temperature in this room is fine | 这个房间的温度适宜 |

the HCQ-P's reliable measurement of comfort over time. Detailed results are presented in Table 4.

### Content validity

In terms of content validity, all experts unanimously agreed that every item of the HCQ-P was relevant to the measurement objectives. Consequently, both the I-CVI and the S-CVI achieved the maximum value of 1, indicating perfect agreement among the experts regarding the questionnaire's content validity.

### Construct validity

Figure 2 presented the CFA model for the HCQ-P, illustrating the factor loadings for each item. After removing item 26 (I would like to see my doctor more often), item 3 (I have enough privacy), item 7 (My beliefs give me peace of mind), item 29 (I can raise above my pain) and item 41 (My God is helping me), the model fit indices ($\chi^2 = 3{,}256.394$, $df = 884$, $\chi^2/df = 3.684 < 5$) confirmed the validity of the four-factor model, ensuring its suitability for use in Chinese hospice care contexts.

## DISCUSSION

Hospice care, pioneered by *Saunders (1978)*, aims to ensure patients spend their final days in comfort, peace, and dignity while providing psychological and spiritual support to their families and caregivers. This concept has gained widespread acceptance globally, especially in developed countries (*Finkelstein et al., 2022*). In developing countries like China, increased awareness about quality of life has significantly heightened interest in hospice care, although research in this field is still emerging (*Zhong et al., 2024*). The HCQ-P has been validated as an effective and reliable tool for assessing comfort in terminally ill patients (*Novak et al., 2001*; *Kolcaba et al., 2004*; *Lorente, Losilla & Vives, 2018*). Thus, the adaptation and validation of the Chinese version of the HCQ-P are crucial steps towards meeting the demand for culturally adapted hospice care tools in China.

To address the need for culturally appropriate hospice care assessment instruments, we translated and adapted the HCQ-P into Chinese and rigorously evaluated its psychometric properties. During the translation process, some ambiguous item expressions in the Chinese

**Table 2  Demographic characteristics of the sample.**

| Characteristics | n | % |
|---|---|---|
| **Gender** | | |
| Male | 186 | 48 |
| Female | 174 | 52 |
| **Age (years)** | | |
| 19–45 | 32 | 9 |
| 46–60 | 107 | 30 |
| 61–75 | 187 | 52 |
| 76–102 | 34 | 9 |
| **Educational level** | | |
| Junior high school or below | 103 | 29 |
| High/Vocational high/Technical secondary school | 121 | 34 |
| Associate degree/Bachelor's degree | 118 | 33 |
| Master's degree or above | 18 | 5 |
| **Marital status** | | |
| Married | 279 | 78 |
| Single | 18 | 5 |
| Divorced | 31 | 9 |
| Widowed | 32 | 9 |
| **Medical expenses** | | |
| Basic medical insurance | 190 | 53 |
| Public medical care | 93 | 26 |
| Cooperative medical care | 58 | 16 |
| Out-of-pocket | 19 | 5 |
| **Disease diagnosis** | | |
| Lung cancer | 91 | 25 |
| Stomach cancer | 16 | 4 |
| Colon cancer | 54 | 15 |
| Liver cancer | 7 | 2 |
| Esophageal cancer | 20 | 6 |
| Pancreatic cancer | 8 | 2 |
| Lymphoma | 14 | 4 |
| Ovarian cancer | 22 | 6 |
| Breast cancer | 18 | 5 |
| Nasopharyngeal cancer | 49 | 14 |
| Others | 61 | 17 |
| **Duration of disease** | | |
| ≤1 year | 157 | 44 |
| 1–5 years | 155 | 43 |
| >5 years | 48 | 13 |

**Table 2** (*continued*)

| Characteristics | *n* | % |
|---|---|---|
| **Surgical history** | | |
| No surgery history | 158 | 44 |
| Surgery history | 202 | 56 |
| **Chemotherapy history** | | |
| No chemotherapy history | 75 | 21 |
| Chemotherapy history | 285 | 79 |
| **Radiotherapy history** | | |
| No radiotherapy history | 246 | 68 |
| Radiotherapy history | 114 | 32 |
| **Life education** | | |
| No life education received | 237 | 66 |
| Life education received | 123 | 34 |
| **Type of previous occupation** | | |
| Manager | 55 | 15 |
| Technician | 95 | 26 |
| Staff | 117 | 33 |
| Farmer | 57 | 16 |
| Unemployed | 36 | 10 |

context were identified and revised after consultation with experts. These adjustments ensured that the wording accurately conveyed the intended meaning and aligned with cultural and linguistic nuances, thereby enhancing the clarity and applicability of the HCQ-P in Chinese hospice care settings. The findings confirm the HCQ-P we translated exhibits sufficient validity (content validity and construct validity), satisfactory reliability (internal consistency and test-retest reliability), and no floor or ceiling effects.

The high internal consistency of the Chinese version of the HCQ-P was evidenced by an overall Cronbach's α of 0.94, with Cronbach's α values for all four comfort dimensions exceeding 0.7. The test-retest reliability, reflected by an overall ICC of 0.93, further confirmed the questionnaire's stability for repeated assessments. Notably, item 26 ('I would like to see my doctor more often') was removed due to its item-total correlation coefficients failing to reach statistical significance, which subsequently increased the reliability of the Sociocultural Comfort dimension from 0.772 to 0.817. This may be due to the lack of participants in the sample from community health service centers actively involved in hospice care. Above all, these findings align with the strong reliability demonstrated in the original English version (*Novak et al., 2001*) and the cross-cultural adaption,like Portuguese adaptation (*Pinto et al., 2016*) and South Korea (*Kim & Kwon, 2007*), supporting the robustness of the HCQ-P across different cultural contexts.

Additionally, due to cultural differences between Eastern and Western contexts, the Psychospiritual Comfort does not apply particularly well in CFA. Specifically, item 3 (I have enough privacy), item 7 ('My beliefs give me peace of mind'), item 29 ('I can raise above my pain') and item 41 ('My God is helping me') had lower factor loadings. Specifically, the physiological dimension of item 29 ('I can raise above my pain'), in the

**Table 3  Mean, standard deviation and ICC of items ($n = 360$).**

| HCQ-P items | Mean | SD | Item-total |
|---|---|---|---|
| **Physiological comfort** | | | |
| HCQ-P1 | 4.96 | 1.400 | 0.590** |
| HCQ-P2 | 4.62 | 1.823 | 0.593** |
| HCQ-P5 | 4.86 | 1.636 | 0.737** |
| HCQ-P12 | 4.54 | 1.785 | 0.769** |
| HCQ-P14 | 4.49 | 1.773 | 0.819** |
| HCQ-P16 | 4.84 | 1.530 | 0.537** |
| HCQ-P19 | 4.72 | 1.679 | 0.784** |
| HCQ-P27 | 4.12 | 1.841 | 0.664** |
| HCQ-P29 | 4.20 | 1.790 | 0.204** |
| HCQ-P31 | 4.31 | 1.618 | 0.434** |
| HCQ-P34 | 4.53 | 1.650 | 0.720** |
| **Psychospiritual comfort** | | | |
| HCQ-P3 | 4.56 | 1.658 | 0.384** |
| HCQ-P7 | 3.87 | 2.113 | 0.466** |
| HCQ-P9 | 5.31 | 1.248 | 0.615** |
| HCQ-P15 | 4.92 | 1.489 | 0.609** |
| HCQ-P17 | 5.03 | 1.486 | 0.612** |
| HCQ-P22 | 4.42 | 1.839 | 0.739** |
| HCQ-P35 | 5.03 | 1.288 | 0.641** |
| HCQ-P36 | 5.15 | 1.286 | 0.534** |
| HCQ-P40 | 4.65 | 1.594 | 0.742** |
| HCQ-P41 | 3.78 | 1.884 | 0.375** |
| HCQ-P43 | 4.81 | 1.541 | 0.669** |
| HCQ-P45 | 4.93 | 1.471 | 0.667** |
| HCQ-P46 | 4.85 | 1.395 | 0.615** |
| **Sociocultural comfort** | | | |
| HCQ-P4 | 5.35 | 1.119 | 0.668** |
| HCQ-P6 | 3.41 | 1.960 | 0.272** |
| HCQ-P8 | 5.19 | 1.183 | 0.663** |
| HCQ-P10 | 5.60 | 0.862 | 0.699** |
| HCQ-P13 | 4.69 | 1.704 | 0.524** |
| HCQ-P20 | 5.35 | 1.129 | 0.651** |
| HCQ-P23 | 5.27 | 1.263 | 0.632** |
| HCQ-P24 | 3.79 | 2.026 | 0.430** |
| *HCQ-P26* | *2.62* | *1.676* | *−0.027* |
| HCQ-P28 | 5.18 | 1.168 | 0.638** |
| HCQ-P37 | 5.16 | 1.214 | 0.691** |
| HCQ-P39 | 2.01 | 1.319 | −0.216** |
| HCQ-P44 | 5.21 | 1.130 | 0.695** |
| HCQ-P47 | 5.15 | 1.284 | 0.673** |

**Table 3** (*continued*)

| HCQ-P items | Mean | SD | Item-total |
|---|---|---|---|
| HCQ-P48 | 4.83 | 1.636 | 0.525** |
| **Environmental comfort** | | | |
| HCQ-P11 | 5.01 | 1.438 | 0.615** |
| HCQ-P18 | 3.91 | 1.890 | 0.601** |
| HCQ-P21 | 4.91 | 1.636 | 0.643** |
| HCQ-P25 | 1.79 | 1.316 | −0.201** |
| HCQ-P30 | 4.66 | 1.561 | 0.780** |
| HCQ-P32 | 4.45 | 1.661 | 0.731** |
| HCQ-P33 | 4.54 | 1.609 | 0.767** |
| HCQ-P38 | 4.60 | 1.622 | 0.796** |
| HCQ-P42 | 4.88 | 1.340 | 0.679** |
| HCQ-P49 | 5.05 | 1.239 | 0.647** |

**Notes.**

**P < 0.01.

Item-total, Item–total correlation coefficients; HCQ-P, Hospice Comfort Questionnaire-Patients; SD, standard deviation.
Item–total correlation coefficients using Pearson's correlation test; Items that are deleted are displayed in italic.

**Table 4  The scores and reliability ($n = 360$).**

| The HCQ-P | Mean | SD | Cronbach's $\alpha$ | ICC |
|---|---|---|---|---|
| Physical comfort | 50.20 | 11.55 | 0.84 | 0.812*** |
| Psychospiritual comfort | 61.31 | 11.90 | 0.84 | 0.936*** |
| Sociocultural comfort | 66.80 | 10.01 | 0.77 | 0.926*** |
| Environmental comfort | 43.79 | 9.51 | 0.82 | 0.949*** |
| The HCQ-P | 224.12 | 38.421 | 0.94 | 0.930*** |

**Notes.**

***$P < 0.0001$.

HCQ-P, Hospice Comfort Questionnaire-Patients; SD, standard deviation; ICC, intra-class correlation coefficient.

Chinese context, patients may be reluctant to express pain because of cultural habits, especially if they do not want to burden their family members. Therefore, deleting item 29 can reduce the influence of cultural and psychospiritual factors. Item 3 ('I have enough privacy'), item 7 ('My beliefs give me peace of mind') and item 41 ('My God is helping me') in psychospiritual comfort exhibit relatively low factor loadings. This may be due to cultural differences between Eastern and Western contexts. In Western cultures, privacy is often considered a fundamental aspect of comfort, whereas in some Eastern cultures, the concept of privacy might be less emphasized in the same context (*Whitman, 2003*). Similarly, God may resonate differently across cultures, particularly when considering the varying religious beliefs and spiritual practices prevalent in different cultural settings (*Wikan, 1992*). Despite this, the overall structural and the other content validity of the HCQ-P was supported by unanimous expert approval during the content validity assessment and significant factor loadings in the CFA after item adjustments. These outcomes indicate that the HCQ-P is not only statistically robust but also resonates well with the cultural and clinical contexts of Chinese hospice care.

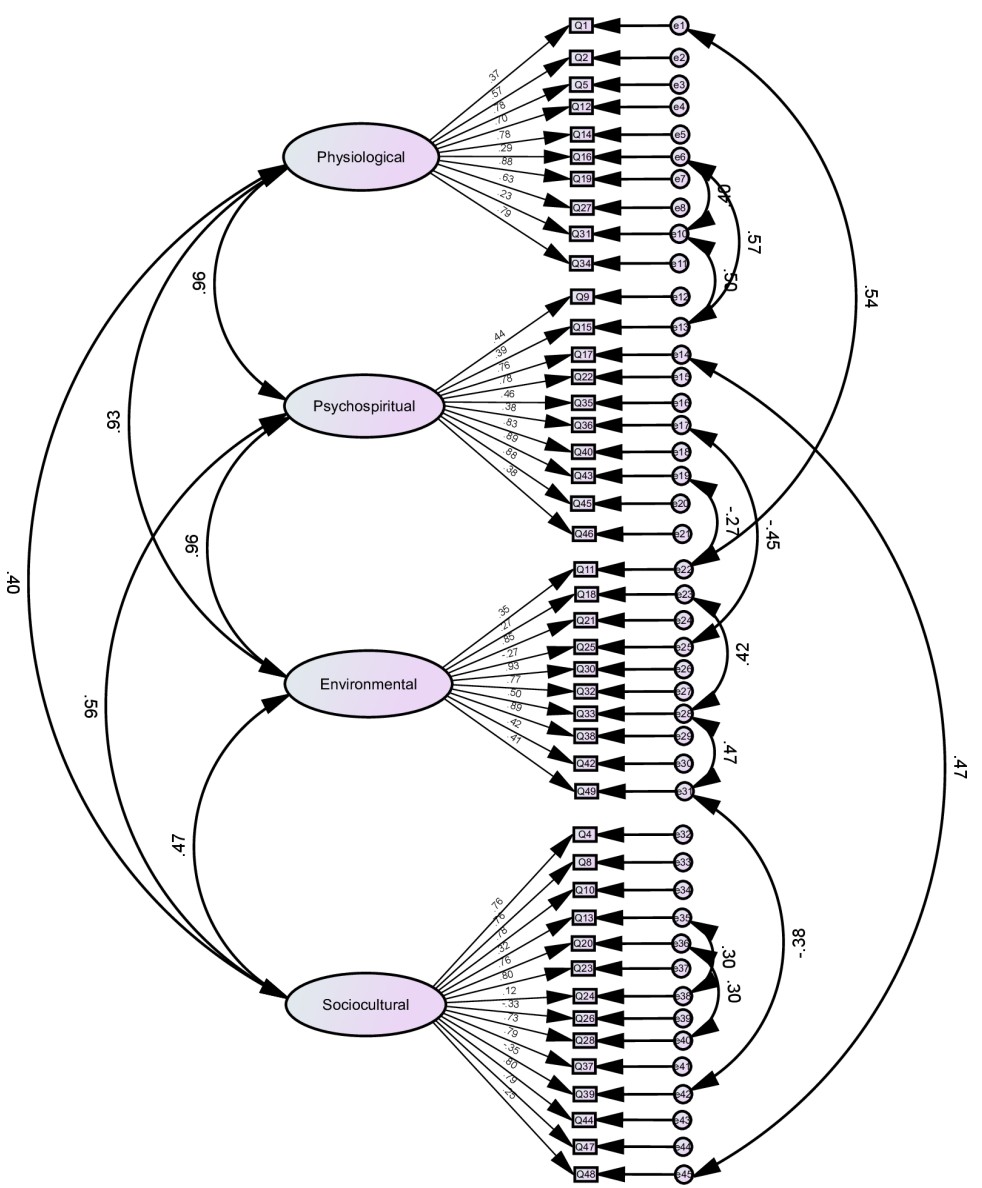

**Figure 2 Confirmatory factor analysis.**

In summary, this research's strengths lie in its rigorous methodology and strict adherence to international guidelines during the translation and cross-cultural adaptation of the HCQ-P, ensuring robustness and scientific validity. However, it does face limitations. First, the sample size was constrained by time and financial limitations, which may affect generalizability. Additionally, the reliance on self-reported data introduces bias, and the sample, largely composed of hospitalized patients, may not represent other hospice populations, such as home-based care recipients. To address these issues, future studies should involve larger, more diverse samples using random sampling across various healthcare settings, including home-based hospice care, to improve external validity.

Furthermore, while some items were removed due to low factor loadings, these could still be relevant for certain patients. Future research should explore alternative wordings through qualitative interviews to refine the scale for cultural relevance. Lastly, in the futurer, the HCQ-P's performance would be evaluated in different disease groups, and comparisons with tools like the General Comfort Questionnaire (GCQ) could help identify strengths and areas for improvement.

## CONCLUSION

We have translated and culturally adapted the Hospice Comfort Questionnaire-Patients (HCQ-P) into Chinese. Psychometric assessments confirm that the Chinese version is both reliable and valid for evaluating hospice care practices in China. Given the constrained sample size of this initial study, further research with a larger and more diverse group of hospice care patients is recommended to enhance the robustness of the findings. Overall, this research equips Chinese healthcare providers and administrators with a validated instrument to assess and address patients' comfort in hospice care, potentially facilitating improvements in service delivery. Additionally, the HCQ-P scores can be utilized to gauge the effectiveness of hospice care training programs or policy implementations, contributing to the broader development of hospice care in China.

### Funding
The authors received no funding for this work.

### Competing Interests
The authors declare there are no competing interests.

### Author Contributions
- Nana Xu conceived and designed the experiments, performed the experiments, analyzed the data, prepared figures and/or tables, authored or reviewed drafts of the article, and approved the final draft.
- Xu Yan conceived and designed the experiments, performed the experiments, authored or reviewed drafts of the article, and approved the final draft.
- Xiaohong Ou conceived and designed the experiments, performed the experiments, authored or reviewed drafts of the article, and approved the final draft.
- Jun Ren conceived and designed the experiments, performed the experiments, authored or reviewed drafts of the article, and approved the final draft.
- Qiaoqin Wan conceived and designed the experiments, performed the experiments, analyzed the data, prepared figures and/or tables, authored or reviewed drafts of the article, and approved the final draft.

### Human Ethics
The following information was supplied relating to ethical approvals (i.e., approving body and any reference numbers):

This study was obtained the ethical approval from the Institutional Review Board of Peking University (Approval No. IRB00001052-24038).

## Data Availability

The raw measurements of patients are available in the Supplementary Files. These participants were used for cross-sectional survey in psychometric evaluation.

## Supplemental Information

Supplemental information for this article can be found online at http://dx.doi.org/10.7717/peerj.19562#supplemental-information.

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
