# Peer review of "Adaptation and validation of the Chinese version of the Hospice Comfort Questionnaire-Patient (HCQ-P)"

_PeerJ, doi:10.7717/peerj.19562_

## Round 0.1 · original submission · Minor Revisions

Please address the reviewer comments.

Reviewer 1 ·

Basic reporting

Strength
The manuscript is well-structured, adhering to the standard format expected in scientific publications. It includes clear sections such as Introduction, Methods, Results, Discussion, and Conclusion. The authors provide a comprehensive background on hospice care and the importance of culturally adapted tools like the HCQ-P, which establishes the context for their study.

Weaknesses and Suggestions
i.Language and Grammar Issues
While the manuscript is generally well-written, there are minor grammatical inconsistencies and awkward phrasing in some sections.

Line 45: "Hospice care is a patient-centered model aimed at improving comfort and preserving dignity at the end of life." This sentence could be rephrased for clarity: "Hospice care is a patient-centered approach that aims to enhance comfort and preserve dignity during the end-of-life period."

Line 108: "Experts inquiry" appears to be a typographical error and should be corrected to "Experts' inquiry."

Suggestion : A thorough proofread by a professional editor or native English speaker would improve readability and ensure consistency.

ii.Figures and Tables
Figure 2 (Confirmatory Factor Analysis) lacks detailed labeling and explanation in the caption. Readers unfamiliar with CFA may find it difficult to interpret without additional guidance.

Suggestion : Include a brief description of the figure’s key components (e.g., factor loadings, model fit indices) in the caption or main text.

iii.Raw Data Availability
The raw data files mentioned in the submission guidelines are not discussed in the manuscript. Transparency about data availability is crucial for reproducibility.

Suggestion : Add a statement in the Methods section confirming that raw data will be made available upon request or through an open-access repository.

Experimental design

Strengths
The study follows a rigorous two-phase process: translation/cultural adaptation and psychometric evaluation, aligning with international guidelines for cross-cultural adaptation of instruments.The use of expert consultations, cognitive testing, and statistical analyses (Cronbach’s α, ICC, CFA) demonstrates a robust methodology.

Weaknesses and Suggestions
i.Sampling Methodology
The authors used convenience sampling, which limits the generalizability of the findings. As noted in the Discussion (Lines 336–341), this approach introduces potential bias.

Suggestion : Future studies should employ random sampling across diverse regions and healthcare settings (e.g., community health service centers) to enhance external validity.

ii.Sample Size Justification
Although the sample size calculation is explained using the Kendall method, the rationale for increasing the sample by 20% to account for non-responses is not fully justified.

Suggestion : Provide a clearer explanation of how the 20% adjustment was determined and whether sensitivity analyses were conducted to confirm its adequacy.
iii.Missing Items in Psychometric Evaluation
Items HCQ-P3, HCQ-P7, HCQ-P29, and HCQ-P41 were removed due to low factor loadings, but the justification for these exclusions could be expanded.

Suggestion : Discuss why these items might perform differently in the Chinese cultural context and explore whether alternative wording could retain them in future iterations.

Validity of the findings

Strengths
The results demonstrate strong psychometric properties, including high Cronbach’s α (0.94) and ICC (0.93), indicating excellent reliability. Content validity was unanimously supported by experts, with both I-CVI and S-CVI reaching the maximum value of 1.

Weaknesses and Suggestions
i.Construct Validity Concerns :
The removal of four items from the Psychospiritual Comfort dimension raises questions about the cultural relevance of this subscale. Specifically, privacy, beliefs, and spirituality may not resonate as strongly in the Chinese context compared to Western cultures.

Suggestion : Conduct qualitative interviews with patients and caregivers to better understand how these concepts are perceived in Chinese hospice care. This could inform refinements to the scale.

ii.Floor and Ceiling Effects
While no significant floor or ceiling effects were reported, only two participants scored at the extremes (Lines 227–228). This small number makes it difficult to draw definitive conclusions.

Suggestion : Increase the sample size in future studies to confirm the absence of floor/ceiling effects.

iii.Test-Retest Reliability
The test-retest reliability was assessed with only 20 participants, which is relatively low for robust ICC calculations.
Suggestion : Expand the retest sample size to ensure more reliable estimates of stability over time.

Additional comments

Strengths
The study addresses a critical gap in hospice care assessment tools for the Chinese population, contributing significantly to the field. The authors have successfully translated and culturally adapted the HCQ-P, ensuring its applicability in Chinese clinical settings.

Weaknesses and Suggestions
i.Discussion Section
The discussion is informative but could benefit from a deeper exploration of the implications of removing certain items (e.g., HCQ-P29: “I can raise above my pain”). How does this impact the overall utility of the scale?

Suggestion : Elaborate on the practical consequences of excluding these items and propose strategies for addressing similar challenges in future adaptations.

ii.Limitations
The limitations section (Lines 336–341) acknowledges biases inherent in self-reported data and convenience sampling but does not address potential issues related to the homogeneity of the sample (e.g., most participants were hospitalized patients).

Suggestion : Highlight the need for validation in other populations, such as home-based hospice care recipients.

iii.Future Research Directions
The conclusion briefly mentions the need for further research with larger samples but does not specify other areas for exploration.

Suggestion : Suggest investigating the HCQ-P’s performance in different disease groups or comparing it with existing tools like the GCQ.

Reviewer 2 ·

Basic reporting

In this manuscript, the authors introduced the Hospice Comfort Questionnaire-Patients (HCQ-P) to China, conducting translation, cultural adaptation, and psychometric validation. In general, this manuscript is well-written, with a clear structure and no significant typographical or grammatical errors, which enhances readability. The figures and tables are well-designed and effectively complement the text.

The author conducted comprehensive literature review of the current hospice care situation in China, the available comfort assessment tools for hospice care, and the need to introduce HCQ to China to fill the gap for comprehensive hospice care assessment.
Minor comments: in line 16: typo “corss-cultural”. In line 50, please add brackets for “Ambuel et al.”.

Experimental design

The authors clearly define the major purpose of this paper: translation and cross-cultural adaptation of the HCQ-P into Chinese and evaluation of the psychometric properties. Multiple rounds of translations, cultural adaptions and evaluations ensure the accuracy in creating the Chinese version of HCQ-P. And the evaluations of HCQ-P in terms of content and construct validity and internal consistency and test-retest reliability are comprehensive and align well with typical process of questionnaire evaluation.
One minor comment: in Line 149 please provide reference for convenience sampling method.

Validity of the findings

Based on the rigorous study design, the authors proved high internal consistency and test-retest reliability of the Chinese version of HCQ-P based on data collected from 360 hospice care patients.

---

## Round 0.2 · accepted · Accept

Thank you for addressing all of the reviewer comments. Your study is now ready for publication - congratulations!

Reviewer 2 ·

Basic reporting

The authors have addressed all the comments.

Experimental design

The authors have addressed all the comments.

Validity of the findings

The authors have addressed all the comments.